# Regnase-1-mediated regulation of neutrophils modulates SARS-CoV-2 pneumonia

Keiko Yasuda[1,2☯*], Junichi Aoki[1,3☯], Kotaro Tanaka[1], Shintaro Shichinohe[4], Chikako Ono[5,6,7], Alexis Vandenbon[8], Daiya Ohara[9], Yukiko Muramoto[10], Songling Li[11], Daisuke Motooka[12], Hitomi Watanabe[9], Keiji Hirota[9], Gen Kondoh[9], Takeshi Noda[10], Daron M. Standley[11], Yuzuru Ikehara[13], Seiji Okada[3], Tokiko Watanabe[4,6,7], Yoshiharu Matsuura[5,6,7], Osamu Takeuchi[1*]

1 Department of Medical Chemistry, Graduate School of Medicine, Kyoto University, Kyoto, Japan, 2 Department of Immunology, Nagoya City University Graduate School of Medical Sciences, Nagoya, Japan, 3 Department of Orthopaedic Surgery, Graduate School of Medicine, The University of Osaka, Suita, Japan, 4 Department of Molecular Virology, Research Institute for Microbial Diseases, The University of Osaka, Suita, Japan, 5 Laboratory of Virus Control, Research Institute for Microbial Diseases, The University of Osaka, Suita, Japan, 6 Center for Infectious Disease Education and Research, The University of Osaka, Suita, Japan, 7 Center for Advanced Modalities and DDS, The University of Osaka, Suita, Japan, 8 Laboratory of Tissue Homeostasis, Institute for Life and Medical Sciences, Kyoto University, Kyoto, Japan, 9 Laboratory of Integrative Biological Science, Institute for Life and Medical Sciences, Kyoto University, Kyoto, Japan, 10 Laboratory of Ultrastructural Virology, Institute for Life and Medical Sciences, Kyoto University, Kyoto, Japan, 11 Department of Genome Informatics, Research Institute for Microbial Diseases, The University of Osaka, Suita, Japan, 12 NGS core facility, Bioinformatics Center, Research Institute for Microbial Diseases, The University of Osaka, Suita, Japan, 13 Department of Pathology, Graduate School of Medicine, Chiba University, Chiba, Japan

☯ These authors contributed equally to this work.
* k-yasuda@med.nagoya-cu.ac.jp (KY); otake@mfour.med.kyoto-u.ac.jp (OT)

## Abstract

The innate immune response to viral infection needs to be tightly regulated to ensure effective pathogen clearance while avoiding excessive immune activation. During SARS-CoV-2 infection, however, the immune system often fails to elicit appropriate responses, resulting in cytokine-release syndrome in patients with COVID-19. In this study, we show that reduced expression of Regnase-1, an RNase that negatively regulates immune cell activation, confers resistance to infection with the mouse-adapted SARS-CoV-2 MA10 strain. In *Regnase-1*[+/–] mice, altered neutrophil function contributed to the amelioration of MA10-induced pneumonia. Single-cell RNA sequencing of lung tissue during MA10 infection revealed four distinct neutrophil subsets, and among these, a subset characterized by an interferon-stimulated gene (ISG) signature was decreased in *Regnase-1*[+/–] mice. Furthermore, *Regnase-1*[+/–] neutrophils exhibited reduced ISG expression without corresponding changes in proinflammatory gene expression. Regnase-1 was found to repress the expression of *Tsc22d3*, a gene involved in the negative regulation of interferon responses, through its 3′ untranslated region. Collectively, these findings suggest that Regnase-1 attenuates

**Data availability statement:** ScRNA-seq and RNA-seq data sets are available under the accession numbers DRR609866-DRR609879 in DNA Data Bank of Japan (https://ddbj.nig.ac.jp/search/entry/bioproject/PRJDB18958). All other data associated with this manuscript are included in the figures and Supporting information.

**Funding:** This work was supported by JSPS (Japan Society for the Promotion of Science) KAKENHI Grants-in-Aid for Scientific Research (JP23H00402, 25H01318, K.Y., O.T., T.W. and S.S.), JSPS Core-to-Core Program (A. Advanced Research Networks) (JPJSCCA20240006, K.Y. and O.T.), JST (Japan Science and Technology Agency) Moonshot (JPMJMS2025, K.Y. and O.T.), AMED (Japan Agency for Medical Research and Development) Research Program on Emerging and Re-emerging Infectious Diseases (25ae0121030, K.Y., O.T. and T.W.), AMED Advanced Research and Development Programs for Medical Innovation (AMED-CREST) (JP22gm1610010, T.W.), JSI (Japanese Society for Immunology) Research Support Program for Young Women Researchers (K.Y.), BIKEN Taniguchi Scholarship (K.T.), Kyoto University SARS-CoV-2 Infectious Disease Control Research Support Grant (K.T.) and Takeda Science Foundation (T.W.). The funders had no role in study design, data collection and analysis, decision to publish, or preparation of the manuscript.

**Competing interests:** The authors have declared that no competing interests exist.

resistance to SARS-CoV-2 MA10 infection by promoting excessive interferon responses in neutrophils.

---

## Author summary

Significant progress has been made in understanding the pathophysiology of SARS-CoV-2 infection. However, the mechanisms by which innate immune cells are activated to eliminate viruses while restraining hyperinflammation remain unclear. Antiviral innate immune responses are regulated by multiple check-points that balance effective inflammatory activity with the prevention of cytokine release syndrome. In this study, we demonstrate that the expression level of Regnase-1, a posttranscriptional regulator of immune cells, determines resistance to infection with the mouse-adapted SARS-CoV-2 MA10 strain. Reduced Regnase-1 expression in neutrophils significantly enhances resistance to viral infection. A decreased representation of a neutrophil subset characterized by a robust interferon-stimulated gene signature is associated with the attenuation of excessive inflammation. These excessive interferon responses may contribute to increased mortality from a lethal dose of the MA10 strain. Regnase-1 suppresses the expression of *Tsc22d3*, a negative regulator of IFN signaling. This implicates Regnase-1 in the dysregulation of immune reactions during infection. Our findings identify Regnase-1 as a novel regulator of neutrophil function and antiviral immunity, which modulates disease severity and pulmonary inflammation.

## Introduction

The clinical course of COVID-19 exhibits considerable variability. While most patients experience mild to moderate symptoms, 10% to 20% develop pneumonia and severe, life-threatening complications [1]. A hallmark of severe COVID-19 is the excessive production of proinflammatory cytokines, triggered by the recognition of SARS-CoV-2 by pattern recognition receptors [2–7] along with hypercoagulation [8–10]. In addition, accumulating evidence suggests that multiple factors contribute to disease severity, including defects in the type I interferon (IFN) response [11,12]. In contrast, delayed and excessive type I IFN responses may exacerbate inflammation and contribute to the severe progression of COVID-19 [13].

Both innate and adaptive immune cells play critical roles in mediating protective immune responses while also contributing to detrimental inflammation during viral infection. Among these, neutrophils, constituting the most abundant nucleated cells in the blood, serve as a first line of defense against invading microorganisms [14]. They exert antimicrobial functions through mechanisms such as degranulation, production of reactive oxygen species, phagocytosis, and the formation of neutrophil extracellular traps (NETs) [15,16]. In patients with severe COVID-19, elevated neutrophil counts and increased neutrophil-to-lymphocyte ratios in peripheral blood are

frequently observed [17–19]. Additionally, dysfunctional low-density neutrophils have been reported in the peripheral blood of severe COVID-19 patients [20,21], suggesting that both neutrophil increase and dysfunction associated to disease severity. The accumulation of neutrophils in the lungs, as documented in autopsy cases of life-threatening COVID-19, along with NET formation [22], provides compelling evidence for the critical role of neutrophils in disease pathogenesis. Nevertheless, the molecular mechanisms regulating neutrophil function during SARS-CoV-2 infection remain poorly understood.

Regnase-1 (Reg1) functions as an RNase that suppresses inflammation by degrading mRNAs encoding inflammation-related genes such as *Il6*, *Il12b* and *Il1b* in immune and non-immune cells [23,24]. Reg1 recognizes stem-loop structures containing a pyrimidine-purine-pyrimidine loop sequence in the 3′ untranslated regions (UTRs) of target mRNAs and degrades them in a translation-dependent manner. Mice deficient in *Reg1* (*Reg1*$^{-/-}$) develop severe inflammatory and autoimmune diseases [23]. Moreover, the Reg1 haploinsufficiency exacerbates autoimmune disease mouse models such as experimental autoimmune encephalomyelitis and imiquimod-induced psoriasis [25–27]. However, the *in vivo* role of Reg1 during viral infection, particularly in the context of SARS-CoV-2 infection, remains uncharacterized.

In this study, we investigated the role of Reg1 on SARS-CoV-2 infection by utilizing a mouse-adapted SARS-CoV-2 strain (MA10), which replicates severe pneumonia observed in life-threatening COVID-19 (ref.[28]). *Reg1*$^{+/-}$ mice exhibited increased resistance to MA10 infection compared to wild-type (WT) mice, depending on the activity of neutrophils. Furthermore, single-cell RNA sequencing analysis suggests that Reg1 modulates both infiltration and functional characteristics of neutrophil subsets in the lung during MA10 infection. Collectively, these findings uncover the roles of Reg1-mediated control of neutrophils in regulating immune responses against SARS-CoV-2 pneumonia.

## Results

### Amelioration of SARS-CoV-2 MA10-induced pneumonia under *Reg1* haploinsufficiency in mice

We investigated the *in vivo* role of Reg1 in mice intranasally infected with the mouse-adapted SARS-CoV-2 (MA10) strain. Whereas *Reg1* haploinsufficient (*+/-*) mice exhibited reduced levels of Reg1 expression [29], immune cell proportions in the lungs remained comparable between *Reg1*$^{+/-}$ and WT mice under steady state conditions (S1A Fig). Following infection with $10^2$ TCID$_{50}$ of MA10, *Reg1*$^{+/-}$ mice showed significantly less weight loss compared to WT mice which lost approximately 10% of their body weight and required 10 days for full recovery (Fig 1A). Histopathological analysis revealed that bronchial damage, alveolar obstruction, and airway lumen retention were less severe in *Reg1*$^{+/-}$ mice compared to WT mice at 3 and 6 days post infection (dpi) (Fig 1B). By contrast, intranasal PBS treatment did not induce pathological lesions in the lungs from either *Reg1*$^{+/-}$ or WT mice (S1B Fig), indicating that observed changes were attributable to MA10 infection. Moreover, diffuse alveolar damage (DAD) scores, assessed based on cellular sloughing, hyalinization, and necrosis, were significantly lower in *Reg1*$^{+/-}$ mice (Fig 1B). Furthermore, infection with higher dose ($10^3$ TCID$_{50}$) of MA10, which resulted in 100% mortality in WT mice by 6 dpi, did not lead to any fatalities in *Reg1*$^{+/-}$ mice. Instead, these mice began to gain body weight by 5 dpi after an initial approximately 10% weight loss (Fig 1C). Consistently, immune cell infiltration and DAD scores were significantly lower in *Reg1*$^{+/-}$ mice at 3 and 6 days following infection with $10^3$ TCID$_{50}$ of MA10 (Fig 1D).

Examination of the immune cell populations in the lungs revealed increased numbers of neutrophils and alveolar macrophages in *Reg1*$^{+/-}$ mice compared with WT mice at 3 dpi with $10^2$ TCID$_{50}$ of MA10 (Fig 1E). However, at 6 dpi, the number of alveolar macrophages had decreased in both WT and *Reg1*$^{+/-}$ mice. Nevertheless, Reg1 haploinsufficiency failed to alter the numbers of other pulmonary innate and acquired immune cells examined (Fig 1E).

Then we examined if Reg1 expression alters cytokine expression during MA10 infection. We observed that *Reg1*$^{+/-}$ mice exhibited increased expression of *Il6* and *Cxcl1* at 6 dpi without altering the expression of other cytokines, such as *Il1b*, *Tnf*, *Il12b*, *Ifnb1* and *Ifna* (Figs 1F and S2A). Nevertheless, the levels of MA10 viral RNA corresponding to the nucleocapsid (*N1*, *N2*) and exonuclease (*ORF1b*) regions were comparable between lungs from WT and *Reg1*$^{+/-}$ mice at both

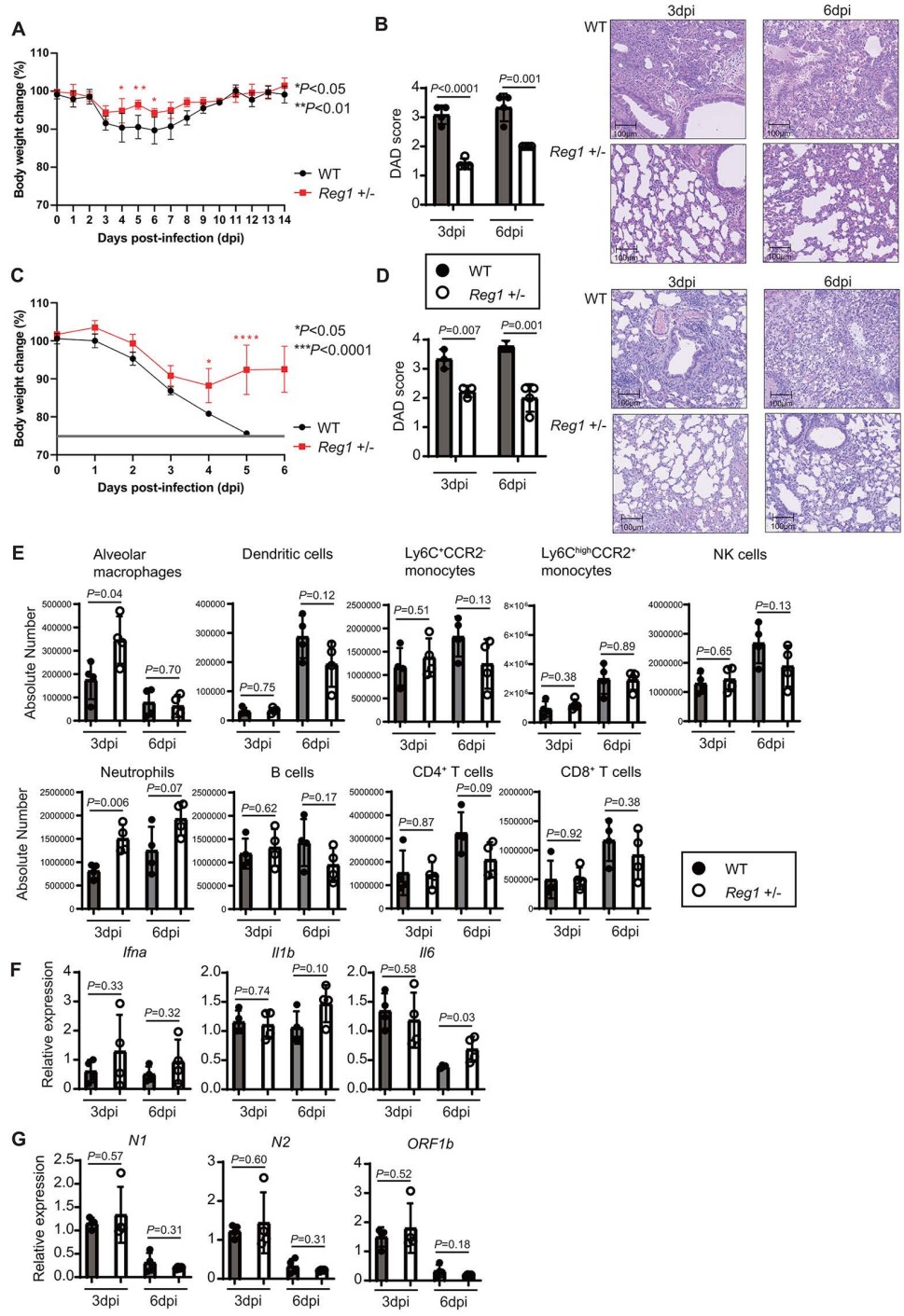

**Fig 1. Ameliorated pneumonia and increased pulmonary neutrophil infiltration in MA10-infected *Reg1*[+/−] mice. A)** Female BALB/c WT and *Reg1*[+/−] mice (12-weeks old, n = 4 per group) were intranasally infected with $10^2$ TCID$_{50}$ MA10. Body weight was monitored daily from -2 to 14 days dpi. The average weight from day -2 to 0 for each group was defined as 100%, and subsequent changes are shown as mean ± s.d. Significant differences was determined by two-way ANOVA with Bonferroni's post-test (*$P < 0.05$, **$P < 0.01$). **B)** Representative H&E-stained lung sections (200 × magnification) from MA10-infected mice collected at 3 and 6 dpi. Lung injury was evaluated in a blinded manner using the DAD scoring at these time points. **C)** Female BALB/c WT and *Reg1*[+/−] mice (12-weeks old, n = 4 per group) were intranasally infected with $10^3$ TCID$_{50}$ MA10. Body weight was monitored daily from -2 to 6 dpi. The average weight from day -2 to 0 was defined as 100%, and changes were shown as the mean ± s.d. Significant difference was assessed by two-way ANOVA with Bonferroni's post-test (*$P < 0.05$, ****$P < 0.0001$). **D)** Representative H&E-stained lung sections (200 × magnification) from

MA10-infected mice collected at 3 and 6 dpi. Lung injury was evaluated using the DAD scoring at 3 dpi (n = 3 per group) and 6 dpi (n = 3 for WT, n = 4 for *Reg1*+/–). E) Absolute numbers of the indicated cell populations infiltrating lungs at 3 and 6 dpi. Cell numbers were calculated by multiplying the total number of cells from lung single cell suspension by the proportion of each population determined by flow cytometry. F) qPCR analysis of *Ifna, Il1b* and *Il6* mRNA expression in RNA extracted from infected lungs at 3 and 6 dpi. G) qPCR analysis of MA10 viral RNA, including *nucleocapsid phosphoprotein* (*N1, N2*) and *ORF1b*, in the lungs of infected mice at 3 and 6 dpi. Bar graphs **(B, D-G)** show the mean ± s.d. (n = 4). *P*-values are shown. (two-tailed unpaired Student's *t*-test).

3 dpi and 6 dpi (Fig 1G), suggesting that *Reg1* haploinsufficiency confers protection against SARS-CoV-2 MA10 infection through mechanisms other than direct viral clearance, potentially involving enhanced accumulation of myeloid cells in the lung.

### Contribution of neutrophils to resistance in *Reg1*+/– mice against MA10 infection

The observation that neutrophil numbers tend to be elevated in *Reg1*+/– mice than in WT mice during MA10 infection, coupled with the higher expression of *Reg1* mRNA in neutrophils among lung-resident immune cells under steady state conditions (Fig 2A), led us to hypothesize that neutrophils contribute to the increased resistance of *Reg1*+/– mice to MA10 infection. To elucidate the role of neutrophils in mediating resistance to MA10 infection under Reg1 haploinsufficiency, we depleted neutrophils in mice using an anti-Ly6G antibody prior to $10^3$ TCID$_{50}$ MA10 infection (Fig 2B and 2C). The Ly6G antibody efficiently depleted neutrophils, but not monocytes, during MA10 infection (Figs 2C and S3A). Whereas both neutrophil-depleted WT and *Reg1*+/– mice exhibited continued weight loss by 5 dpi, there was a significant difference in body weight compared with untreated WT mice at 5 dpi (Fig 2D). Consistently, neutrophil depletion was associated with severe lung inflammation in both WT and *Reg1*+/– mice, although DAD scores did not reach statistically significant compared with WT mice without neutrophil depletion (Fig 2E). DAD scores were similar between neutrophil-depleted WT and *Reg1*+/– mice during MA10 infection (Fig 2E). Notably, neutrophil depletion did not significantly alter the levels of MA10 viral RNAs (*N1, N2* and *Orf1b*) in lungs at 5 dpi in WT and *Reg1*+/– mice (Fig 2F). These results imply that neutrophil depletion does not alter viral clearance. These results indicate that *Reg1*+/– mice no longer displayed enhanced resistance to MA10 infection compared to WT mice in the absence of neutrophils. Collectively, these results suggest that *Reg1* expressed in neutrophils contributes to the regulation of resistance against MA10 infection.

### Reduced number of neutrophils with a high IFN signature in the lungs of MA10-infected *Reg1*+/– mice

To further characterize the phenotypic changes in immune cells during infection with SARS-CoV-2 MA10 strain, we isolated lung CD45+ cells from mice at 0, 3 and 5 days after infection with $10^3$ TCID$_{50}$ MA10 and subjected them to single-cell RNA sequencing (scRNA-seq). We analyzed 7507, 8180 and 12466 cells from WT mice and 4630, 8526 and 11485 cells from *Reg1*+/– mice at 0, 3 and 5 dpi, respectively. Transcriptomic analysis identified 22 distinct cell clusters, allowing for the characterization of lung immune cell populations based on their signature gene expression profiles (Fig 3A). The lethal dose ($10^3$ TCID$_{50}$) MA10 infection induced dynamic alterations in lung immune cell composition, marked by increases in monocytes, macrophages, T and B cells, along with a decrease in alveolar macrophages in both WT and *Reg1*+/– mice (Fig 3B). However, we failed to observe significant differences in the proportions of immune cell subsets including T cells, B cells, NK cells, monocytes/macrophages, alveolar macrophages, neutrophils and plasmacytoid dendritic cells (pDCs) between WT and *Reg1*+/– mice during lethal dose of MA10 infection (Fig 3B). Flow cytometry analysis confirmed that the numbers of immune cells were comparable between WT and *Reg1*+/– mice, except NK cells showing an increase in *Reg1*+/– mice at 6 dpi (S4A Fig). The discrepancy between scRNA-seq and flow cytometry results in NK cells is thought to be due to the difference in sample collection timing at 5 dpi and 6 dpi. Further studies will be required to elucidate the role of Reg1 in NK cells during SARS-CoV-2 MA10 infection. Furthermore, viral RNA levels and cytokine gene expression in the whole lung as well as NK cells, monocytes/macrophages and pDCs were comparable between WT and *Reg1*+/– mice

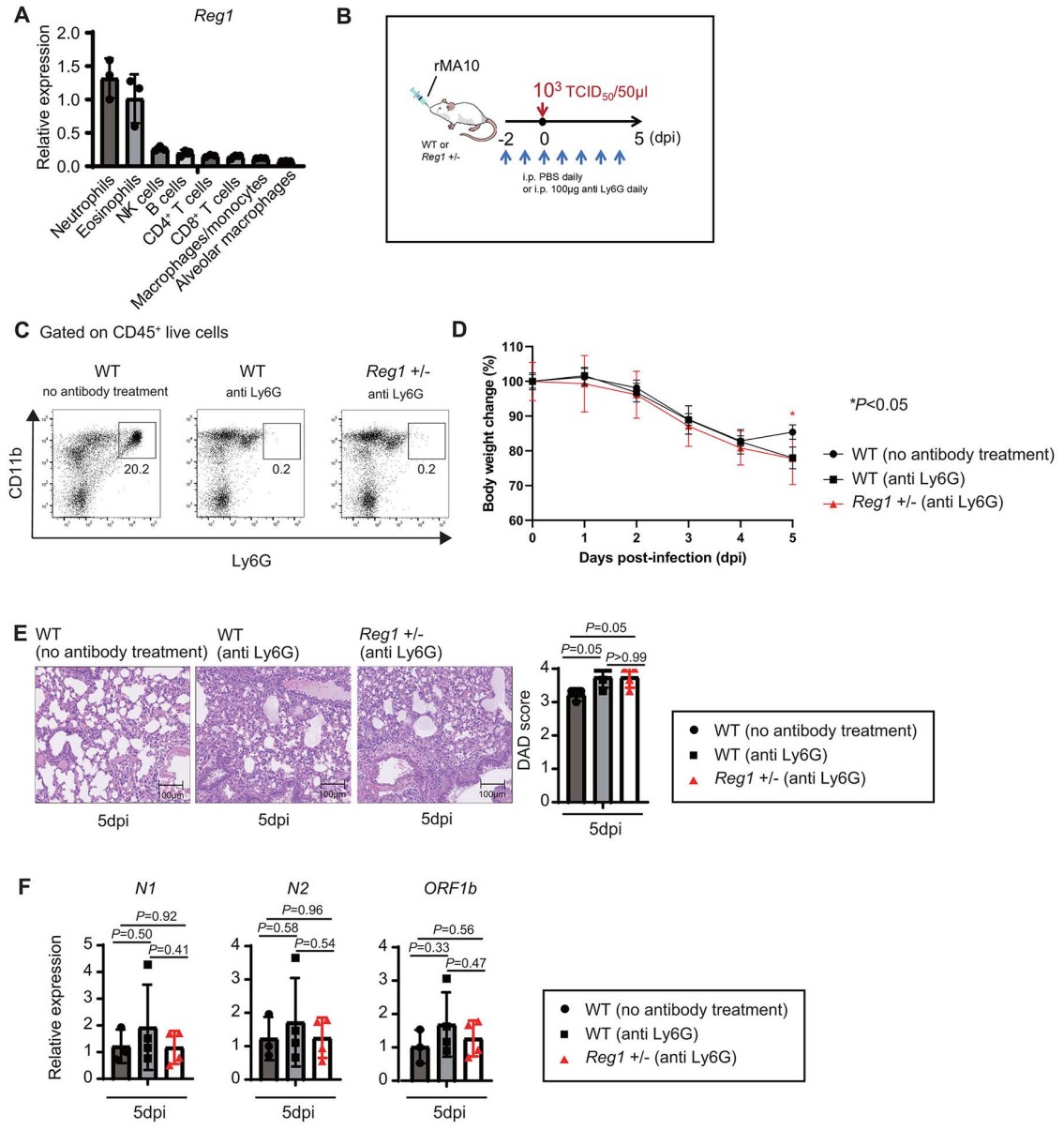

**Fig 2. Neutrophils mediate resistance to MA10 infection in *Reg1*[+/−] mice. A)** qPCR analysis of *Reg1* mRNA expression in the indicated lung cell types under steady state conditions (n = 3). **B)** Schematic representation of antibody treatment and MA10 infection. Female BALB/c WT mice were divided into two groups: no antibody treatment (PBS) group and Ly6G antibody treatment group. Female BALB/c *Reg1*[+/−] mice were also treated with Ly6G antibody. Antibody-treated groups daily received 100 μg/day of Ly6G antibody intraperitoneally (i.p.) until 4 dpi. All mice were intranasally infected with $10^3$ TCID$_{50}$ MA10. Lung-infiltrating cells were collected at 5 dpi for flow cytometry analysis and a portion of the lung tissue was used for histopathological evaluation. **C)** Flow cytometry analysis of neutrophils in CD45$^+$ live cells isolated from infected lung at 5 dpi. **D)** Body weight was monitored from day -2 to 5 dpi. The average weight from day -2 to 0 was defined as 100%, and changes were shown as the mean ± s.d. Significant difference was assessed by two-way ANOVA with Bonferroni's post-test (*$P < 0.05$). **E)** Representative H&E-stained lung sections (200 × magnification) from MA10-infected mice collected at 5 dpi. Lung injury was evaluated using the DAD scoring at 5 dpi (n = 3 for no untreated WT mice, n = 4 for anti Ly6G treated WT and anti-Ly6G treated *Reg1*[+/−]). **F)** qPCR analysis of MA10 viral RNA, including *nucleocapsid phosphoprotein* (*N1*, *N2*) and *ORF1b*, in the lungs of infected mice at 5 dpi. (n = 3 for no untreated WT mice, n = 4 for anti Ly6G treated WT and anti-Ly6G treated *Reg1*[+/−]). The bar graphs **(A, E, F)** show the mean ± s.d. (n = 3 or 4). *P*-values are shown. (two-tailed unpaired Student's t-test).

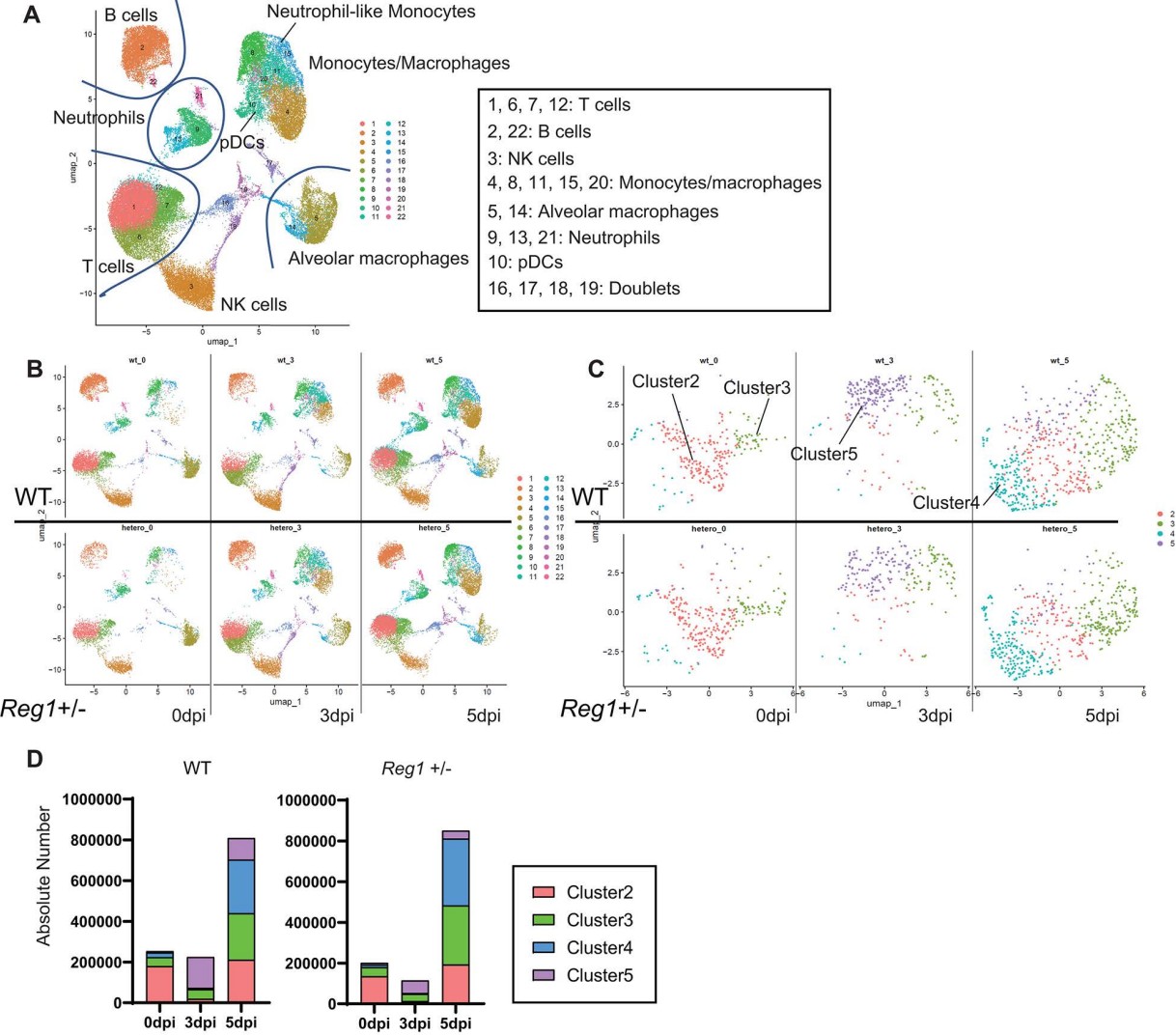

**Fig 3. scRNA-seq reveals the subdivision of lung-infiltrating neutrophils into four clusters following 10³ TCID₅₀ MA10 infection. A)** UMAP representation and clustering of lung infiltrating CD45⁺ cells (0, 3 and 5 dpi, WT mice and *Reg1*⁺/⁻ mice). Data-driven clustering and cell type based on canonical marker genes are shown. **B)** UMAP representation of lung CD45⁺ cells at the indicated time points and genotypes. For each group, the CD45⁺ cells from three mice were combined and processed as a single sample (n = 1) for further analysis. **C)** UMAP representation of lung infiltrating neutrophils, corresponding to clusters 9, 13 and 21 in **(A)**. These neutrophils were further subdivided into four clusters based on gene expression profiles. **D)** Absolute numbers of neutrophils at 0, 3 and 5 dpi in WT mice or *Reg1*⁺/⁻ mice.

(S4B and S4C Fig and S1 Table). These results suggest that the enhanced resistance of *Reg1*⁺/⁻ mice to viral infection is not attributable to altered proinflammatory cytokine production. Furthermore, prior to infection, neutrophils exhibited higher *Reg1* mRNA expression than other lung-resident cell types (Fig 2A). Following infection, neutrophils also maintained higher *Reg1* mRNA expression among innate immune cells (S5A Fig).

Given that neutrophils were shown to contribute to increased resistance to MA10 infection in *Reg1*⁺/⁻ mice (Fig 2D and 2E), we investigated their properties in more detail. scRNA-seq analysis identified eight neutrophil clusters in the lung (S6A Fig). However, based on their gene expression profiles, clusters 1, 6, 7, and 8 likely represented doublets composed of neutrophils and other cells such as T cells or B cells (S6B Fig), and were excluded from further analysis and we

focused on clusters 2–5. Under steady state conditions, a majority of neutrophils belonged to clusters 2, with a smaller population in cluster 3 (Fig 3C and 3D). Following MA10 infection, the proportion of cluster 2 neutrophils initially diminished but reemerged at 5 dpi. In contrast, other neutrophil clusters exhibited distinct kinetics: cluster 5 expanded rapidly but transiently at 3 dpi, whereas clusters 3 and 4 increased notably at 5 dpi. *Reg1* haploinsufficiency had minimal impact on the overall distribution of neutrophil clusters, with the exception of cluster 5, which was significantly reduced at both 3 and 5 dpi (Fig 3C and 3D).

We next characterized the transcriptional features of these neutrophil clusters. Although severe COVID-19 has been associated with emergency myelopoiesis leading to the release of immature neutrophils into circulation [20], neutrophils in clusters 2–5 from both WT and *Reg1*$^{+/-}$ mice expressed genes characteristic of mature neutrophils (e.g., *S100a8*, *S100a9* and *Junb*) and lacked expression of immature markers including *Mpo*, *Elane* and *Gata1* [30–34] (S6C and S6D Fig). These findings indicate that the identified clusters represent functional heterogeneity among mature neutrophils rather than distinct stages of maturation.

Each neutrophil cluster exhibited a distinct gene expression profile (Fig 4A). Clusters 3 and 5 were characterized by a robust, transient induction of ISGs, including *Irf7*, *Ifit1*, *Ifit2*, *Ifit3*, *Rsad2*, and *Isg15*, at 3dpi (Fig 4B). Notably, cluster 3, unlike 5, showed elevated expression of pro-inflammatory cytokine genes and their regulatory elements, including *Il1a, Tnf, Nfkbia* and *Tnfaip3*, in addition to ISGs. In contrast, clusters 2 and 4 exhibited only modest ISG expression. Cluster 4 was enriched for genes involved in neutrophil migration, protease inhibition and genes with unknown functions (e.g., *Cd177*, *Cstdc4*, *Cstdc6*, *Stfa2* and *Stfa3*), whereas cluster 2 lacked prominent transcriptomic features.

The expression of proinflammatory cytokine-associated genes, *Il1a, Tnf, Nfkbia* and *Tnfaip3*, was comparable between WT and *Reg1*$^{+/-}$ neutrophils, or was slightly elevated in *Reg1*$^{+/-}$ mice (Figs 4C and S7A). In contrast, the expression of ISGs including *Ifit1*, *Ifit3*, *Ifitm3*, *Oasl2* and *Usp18* was reduced particularly in clusters 3 and 5 in *Reg1*$^{+/-}$ mice compared to WT, while the expression of *Ifitm3* showed slight increase in cluster 5 of *Reg1*$^{+/-}$ mice at 5 dpi (Figs 4C, 4D, and S7A). Consistently, transcriptome analysis of neutrophils sorted from MA10-infected WT and *Reg1*$^{+/-}$ mice confirmed a downregulation of a gene set associated with type I IFN responses in *Reg1*$^{+/-}$ neutrophils (Fig 4E). Collectively, these findings suggest that *Reg1* haploinsufficiency attenuates type I IFN responses in neutrophils during MA10 infection, which may contribute to the reduced disease severity observed in *Reg1*$^{+/-}$ mice.

## Reg1 suppresses genes including *Tsc22d3* in neutrophils

To investigate the molecular mechanisms by which Reg1 modulates neutrophil functions, we assessed gene expression profiles in lung neutrophils from WT and *Reg1*$^{+/-}$ mice. Given that excessive type I IFN responses in neutrophils may contribute to tissue damage and impair resistance to viral infection, we hypothesized that Reg1 suppresses genes involved in the negative regulation of the IFN responses. RNA-seq analysis of lung neutrophils revealed that Reg1 haploinsufficiency led to the upregulation of 135 genes compared to WT neutrophils (Fig 5A). Among these, the expression of 5 genes reported to be associated with IFN response signature (*Tsc22d3*, *Nr2f2*, *Axl*, *Muc4* and *Fgfr1*) [35–39] were validated by qPCR. Noteworthy, *Tsc22d3* exhibited significant upregulation in *Reg1*-haploinsufficient neutrophils, whereas the other 4 genes did not show significant upregulation (Figs 5B and S8A). Additionally, the expression of *Tsc22d3* in the lung was higher in neutrophils than in other immune cell types in uninfected mice (Fig 5C). To determine whether Reg1 directly regulates *Tsc22d3* mRNA via the 3′ UTR, we conducted the luciferase reporter assays. Overexpression of wild-type Reg1 significantly suppressed the activity of the reporter construct harboring the *Tsc22d3* 3′ UTR (Fig 5D). In contrast, a nuclease-inactive Reg1 mutant (D141N) failed to inhibit the reporter activity (Fig 5D), indicating that Reg1 acts as an RNase to control *Tsc22d3* mRNA expression. These findings suggest that Reg1 modulates the expression of neutrophil genes, including ISG signatures, by degrading mRNA such as *Tsc22d3*.

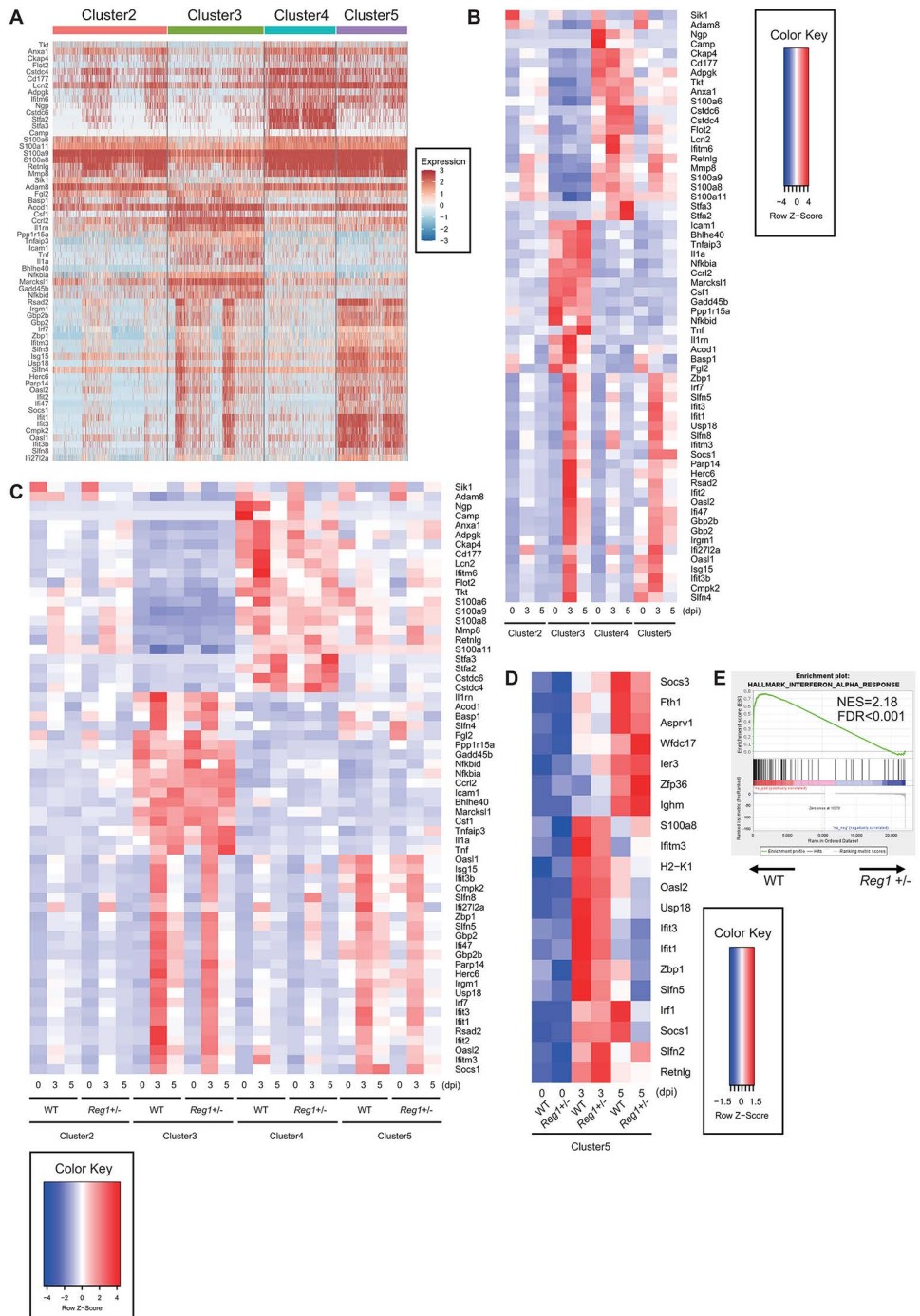

**Fig 4. Attenuated type I IFN signature in lung neutrophils of MA10-infected *Reg1*+/– mice. A)** Heatmap showing differences in gene expression among the four neutrophil subclusters (cluster 2, 3, 4 and 5 from Fig 3C). **B)** Heatmap of the top marker genes ranked by log fold change which are associated with the neutrophil clusters indicated in **(A)**, segmented by time points (0, 3 and 5 dpi). **C)** Heatmap of the top marker genes ranked by log fold change which are associated with the neutrophil clusters indicated in **(A)**, segmented by time points (0, 3 and 5 dpi) and genotype of mice. **D)** Heatmap of genes upregulated in cluster 5 by comparing 0 vs. 3 dpi and 0 vs. 5dpi in WT mice. Gene expression is displayed across time points and genotypes. **E)** GSEA comparing transcriptome profiles of lung-infiltrating neutrophils at 5 dpi ($10^3$ TCID$_{50}$ MA10 infection) between WT and *Reg1*+/– mice using HALLMARK_INTERFERON_ALPHA_RESPONSE gene sets (M5911; genes upregulated in response to IFNα).

PLOS Pathogens

**A**

| Gene_Symbol | Gene_Symbol | Gene_Symbol | Gene_Symbol | Gene_Symbol |
|---|---|---|---|---|
| Skint3 | Clmp | Klhl13 | Ecm1 | Lama2 |
| Dmbt1 | Mn1 | Ptges | Itga7 | Gsn |
| Pax9 | Zfp9 | Ackr3 | Prelp | Lama4 |
| Alox15 | Gm14288 | Loxl2 | Heg1 | Scgb3a2 |
| Cxcl13 | Igfbp3 | Fbn1 | Loxl1 | Cp |
| Npas2 | Lrrc32 | Scara5 | Abca8a | Lcn2 |
| Ltf | Ms4a4d | Heyl | Dpt | Mmp2 |
| Esm1 | Zbtb16 | Gm10184 | Boc | Nid1 |
| Lox | Gm13889 | Slc10a6 | Cygb | Sparcl1 |
| Col5a3 | Vgll3 | Lpar1 | Scn7a | Hyou1 |
| Rem1 | Hsph1 | Chad | Lrg1 | Scd1 |
| Scgb3a1 | Kcna2 | Ubfd1 | Fstl1 | |
| Dio2 | Slco2b1 | Hsd11b1 | Spon1 | |
| Mmp3 | Cebpd | Nrcam | Slit3 | |
| Gfpt2 | Aldh3a1 | Adamtsl2 | Serpinh1 | |
| Arntl | Bpifb1 | Pmepa1 | Mmp14 | |
| Mt2 | Enpp3 | Pdgfra | Gpx3 | |
| Cyp7b1 | Has1 | Cxcr2 | Nbl1 | |
| Ccn2 | Fbln2 | Ndrg2 | Dkk3 | |
| Galnt15 | Pi15 | Chordc1 | Fndc1 | |
| Lgr6 | Flt3l | Manf | Lamb1 | |
| Eln | Dpep1 | Pamr1 | Dcn | |
| Tnc | Slc43a3 | Kirrel | Nr2f2 | |
| Mt1 | Tshz2 | Axin2 | Arrdc3 | |
| Lrat | Jchain | Vcam1 | Osmr | |
| Slc26a4 | Mbd1 | Pigr | Ltbp1 | |
| Htra3 | Lum | Reg3g | Axl | |
| Alpl | Map3k6 | C4b | 1810010H24Rik | |
| Snai2 | Plat | Col5a1 | Muc4 | |
| Tsc22d3 | Zfp970 | Col5a2 | Fgfr1 | |
| Cilp | Adamts15 | Adamts5 | Col6a3 | |

**B**

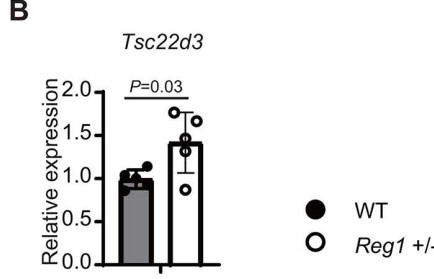

**C**

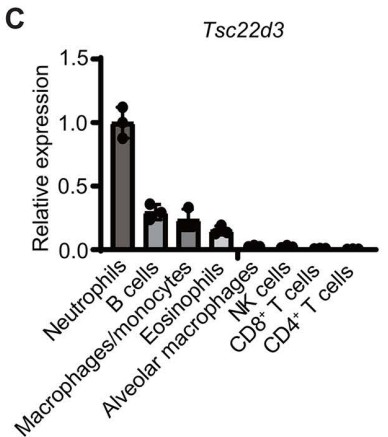

**D**

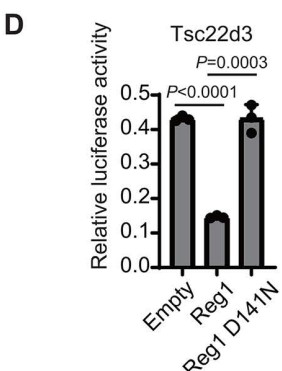

**Fig 5. Reg1 suppresses the expression of genes including *Tsc22d3* via its RNase activity. A)** List of differentially expressed genes upregulated in *Reg1*[+/−] mice, identified by RNA-seq analysis of lung-infiltrating neutrophils from WT and *Reg1*[+/−] mice at 0 dpi (uninfected). The top 135 genes are listed, ranked by the genes with the smallest log2 fold change values (-9.26 ∼ -0.34). Five candidate genes potentially related to neutrophil regulation (*Tsc22d3*, *Nr2f2*, *Axl*, *Muc4* and *Fgfr1*) are highlighted in yellow. Two biological replicates were analyzed. **B)** qPCR of *Tsc22d3* mRNA expression in lung-infiltrating neutrophils at 0 dpi from WT and *Reg1+/−* mice (n = 5 per group). **C)** qPCR analysis of *Tsc22d3* mRNA in the indicated cell types from uninfected lung (n = 3). **D)** Luciferase activity in HEK293T cells transfected with reporter plasmids expressing murine *Tsc22d3* 3' UTR sequence and expression plasmids for Reg1 (WT or D141N) (n = 3 per group). The bar graphs **(B, C, D)** show the mean ± s.d. (n = 3 or 5). *P*-values are shown. (two-tailed unpaired Student's *t*-test).

## Discussion

In the present study, we found that *Reg1*$^{+/-}$ mice exhibited ameliorated pneumonia following infection with the mouse-adapted SARS-CoV-2 MA10 strain compared to WT mice. Neutrophil depletion experiments demonstrated that the reduced *Reg1* expression in neutrophils contributes to the enhanced resistance of *Reg1*$^{+/-}$ mice to infection. scRNA-seq analysis further revealed a decrease of a neutrophil subset with robust IFN signatures under *Reg1* haploinsufficiency during MA10 infection, implicating that excessive type I IFN activation in WT mice is involved in disease pathogenesis. Further, we identified a set of Reg1-suppressed genes in neutrophils including *Tsc22d3*, that may regulate type I IFN signaling. These findings suggest that decreased expression of *Reg1* expression in lung neutrophils provide protective effects against SARS-CoV-2 MA10 infection.

Our experimental model offers several advantages in elucidating the pathogenesis contributing to the severity of SARS-CoV-2 infection. Clinical specimens from COVID-19 patients are typically collected at medical facilities after symptom onset and definitive diagnosis, making it challenging to collect samples from the early stages of infection and track disease progression. In contrast, the SARS-CoV-2 MA10 mouse model employed in this study provided robust evidence for the contribution of neutrophils in resistance to SARS-CoV-2. This model allowed targeted neutrophil depletion and precise temporal control over analysis, enabling a detailed examination of immune cell dynamics in the lung following MA10 infection. An additional advantage of the MA10 mouse model is its suitability for investigating the role of specific genes in COVID-19 resistance using genetically modified mice. In this study, we demonstrated that reduced *Reg1* expression due to haploinsufficiency enhances resistance to MA10 infection, further highlighting the utility of this model for studying host-pathogen interactions.

It should be noted, however, that naturally occurring SARS-CoV-2 variants such as Omicron lineages are also capable of infecting laboratory mice, but generally induce milder disease with limited lung pathology compared with MA10. While this limits their use for modeling life-threatening pneumonia, Omicron-based models may be particularly valuable for investigating host responses associated with upper airway infection, immune evasion, viral persistence, and post-acute or subclinical disease states. Future studies comparing MA10 with Omicron strains in genetically modified mice may therefore help to delineate shared versus strain-specific mechanisms governing neutrophil function, tissue damage, and immune regulation during SARS-CoV-2 infection.

Our neutrophil depletion studies further underscored the critical role of these cells in mediating antiviral resistance. In influenza infection, it has been reported that pneumonia severity worsens in the absence of neutrophils at the time of infection, based on similar neutrophil depletion experiments [40]. We demonstrated that neutrophils are also critical for host defense against MA10 infection. Numerous clinical studies have highlighted the detrimental roles of excessive neutrophil activation in severe COVID-19, including increased peripheral neutrophil counts [1] and elevated neutrophil-to-lymphocyte ratios [41], as well as the formation of NETs, which may induce secondary organ damage, such as thrombus formation [22]. Together with our findings, these observations emphasize the need for tight control of neutrophil activation during viral infection. While neutrophil deficiency impairs antiviral defense, excessive neutrophil activation is a major driver of severe disease and life-threatening complications. Our scRNA-seq analysis further revealed dynamic changes in neutrophil populations during viral infection, providing a detailed characterization of lung immune cells responses. These findings complement prior studies analyzing peripheral blood and bronchoalveolar lavage samples from COVID-19 patients [1,17–21,41–48], offering additional insights into the immunological landscape of SARS-CoV-2 infection.

An adequate IFN response is essential for antiviral host defense. In one hand, both insufficient type I IFN production [42] and the presence of antibodies to type I and type III IFNs [11,12] exacerbate COVID-19. Conversely, delayed yet excessive IFN responses can exacerbate inflammation, contributing to severe disease progression [13]. This dysregulated IFN response, in combination with excessive inflammatory cytokine production, can lead to a "cytokine storm" causing severe lung damage and multiple organ failure [49]. The timing and location of IFN responses are therefore crucial determinants of disease outcomes [13,43–46]. This notion is further supported by studies demonstrating that dexamethasone,

which dampens the ISG response, provides protection against SARS-CoV-2 infection [47,48]. Accordingly, dexamethasone is recommended for treating hospitalized patients suffering from life-threatening COVID-19 requiring oxygen supplementation or mechanical ventilation [50]. Given that *Reg1* haploinsufficiency confers resistance to mouse SARS-CoV-2 infection while attenuating IFN responses, targeted suppression of *Reg1* expression may represent a novel therapeutic strategy for COVID-19 and other severe viral infection.

We observed that the expression of *Tsc22d3* was upregulated, whereas ISG signature genes were suppressed in *Reg1+/−* neutrophils. Furthermore, we demonstrated that Reg1 directly destabilizes *Tsc22d3* via its 3' UTR in an RNase activity-dependent manner. Tsc22d3 (also known as GILZ: glucocorticoid-induced leucine zipper) was originally identified as an anti-inflammatory molecule induced by glucocorticoid [51–53]. A previous study showed that Tsc22d3 suppress ISG upregulation by binding to STAT1 and preventing its nuclear translocation [35]. In addition, Tsc22d3 expression in neutrophils has been shown to exert protective effects in an acute colitis mouse model by inhibiting the MAPK pathway, and thereby preventing excessive neutrophil activation [54]. Thus, increased expression of *Tsc22d3* in *Reg1+/−* neutrophils may confer protection against MA10 infection by damping ISG expression and excessive neutrophil activation. However, further investigations are needed to validate these hypotheses. Moreover, in addition to *Tsc22d3*, Reg1 may target additional mRNAs involved in the regulation of ISG expression in neutrophils. mRNAs upregulated in *Reg1+/−* neutrophils represent the candidates target, and future studies will be required to determine which of these mRNAs are directly degraded by Reg1 and contribute to the control of IFN responses during SARS-CoV-2 MA10 infection.

In this study, we primarily focused on the role of Reg1 expressed in neutrophils in shaping host responses against MA10 infection. However, Reg1 expressed in other cell types may also contribute to disease outcomes. For instance, flow cytometry analysis revealed an increased number of alveolar macrophages in *Reg1+/−* mice compared with WT mice at 3 dpi following low-dose MA10 infection. Future investigations will be required to elucidate the roles of Reg1 in other immune and non-immune cell types during MA10 infection.

It was reported that Regnase-1 can degrade viral RNA in cell systems overexpressing Reg1 [55,56]. However, these studies did not demonstrate a contribution of Reg1-mediated viral RNA degradation to antiviral defense *in vivo*. If Reg1-mediated viral RNA degradation plays a significant role during SARS-CoV-2 infection *in vivo*, one would expect *Reg1+/−* mice to exhibit increased viral loads and exacerbated disease. In contrast, our study showed that *Reg1+/−* mice developed ameliorated SARS-CoV-2 pneumonia, and we did not detect significant differences in viral loads in the lungs. These findings indicate that Reg1 contributes to the exacerbation of SARS-CoV-2 pneumonia *in vivo*, even though we cannot exclude the possibility that Reg1 may degrade SARS-CoV-2 RNA in certain cell types, including airway epithelial cells.

A limitation of this study is the reliance on a mouse model to assess the role of Reg1 in resistance against mouse-adapted SARS-CoV-2 MA10 infection, and its relevance to human COVID-19 remains untested. In addition, in our scRNA-seq analysis, three mice were used per time point and pooled to generate a biological replicate. Owing to this experimental design, statistical comparisons between groups could not be performed, and formal statistical analyses were therefore not feasible for these datasets. Nonetheless, a previous study reported that *Reg1* expression is upregulated in dysfunctional neutrophils in the peripheral blood of patients with severe COVID-19 [20], suggesting that maintaining optimal *Reg1* levels in neutrophils may be protective, consistent with our findings. However, further studies are required to establish the causal relationship between *Reg1* expression and neutrophil functions.

## Materials and methods

### Mice and infections

**Ethics statement.** *Regnase-1−/−* mice on C57BL/6 background [23] were backcrossed more than six times to BALB/c background [29], and *Regnase-1+/−* (BALB/c) mice were used in the present study. WT (BALB/c) mice were purchased from CLEA Japan and bred at the Kyoto University animal facility for at least two weeks before being used in the experiments. All the mice used were maintained under SPF conditions at the animal facility of the Institute for Life and

Medical Sciences, Kyoto University. Twelve-week-old female mice were used for MA10 infection experiments. Mice were infected intranasally with $10^2$ TCID$_{50}$/50μl or $10^3$ TCID$_{50}$/50μl/ MA10 in an approved animal biosafety level 3 facility in Osaka University following regulations. Mouse experiments were approved by the animal experimentation committees of Graduate School of Medicine, Kyoto University and Research Institute for Microbial Diseases, Osaka University, and conducted according to institutional guidelines. Virus-infected mice were euthanized if they lost more than 25% of their initial body weight. Lung tissue and peripheral blood (serum or cells) were collected at the indicated time points for analysis.

**Viruses.** SARS-CoV-2 MA10 was established by a reverse genetics method called CPER (circular polymerase extension reaction) [57] using SARS-CoV-2 strain 2019-nCoV_Japan_TY_WK-5212020 (isolated by NIID) as the backbone. MA10 contains seven mutations, including nsp4: T295I, nsp7: K2R, nsp8: E23G, S: Q493K/Q498Y, P499T, and orf6: F7S, which have been reported as adaptive mutations introduced in SARS-CoV-2 during serial passages in BALB/c mice by Leist et al [28]. All viruses were amplified in VeroE6/TMPRSS2 cells, and the culture supernatants were harvested and titers were measured using the TCID$_{50}$ assay and stored at -80℃ until use. MA10 preparation was done in the biosafety level 3 facility in Research Institute for Microbial Diseases, Osaka University.

**Viral titration of SARS-CoV-2 MA10.** Viral titers were measured by median tissue culture infectious dose (TCID$_{50}$) assays. VeroE6/TMPRSS2 cells (Japanese Collection of Research Bioresources, JCRB1819) were cultured in MEM supplemented with 5% FBS and 1% penicillin/streptomycin and seeded into 96-well cell plates. Samples were serially diluted 10-fold $10^{-1}$ to $10^{-8}$ in the cell culture medium. Dilutions were plated in triplicate on VeroE6/TMPRSS2 cells and incubated at 37℃ for 72 hours. Cytopathic effects were evaluated by microscopy. TCID$_{50}$/ml was calculated by the Reed-Muench method.

## Plasmid construction

Expression plasmids for murine Reg1 and the D141N mutant were as described previously [23]. The 3' UTR sequence of murine Tsc22d3 (796–2135 of NM_001077364.2) was purchased from Thermo Fisher Scientific GeneArt, and was inserted in the pGL3-promoter vector (Promega).

## Luciferase assay

HEK293T cells were transfected with luciferase reporter plasmid pGL3 containing the 3' UTR sequence of murine Tsc22d3 together with expression plasmids for murine Reg1 (WT or D141N) or empty (control) plasmid, and the Relilla luciferase plasmid. Cells were lysed and luciferase activity in the lysates was determined using the Dual-Luciferase Reporter Assay System (Promega) 24 hours after transfection.

## Cell preparation

Lungs were harvested, minced with scissors, and digested in 50μg/ml Liberase (Roche) and 100μg/ml DNase I (Roche) containing RPMI medium for 40 minutes at 37℃. Then EDTA was added to a final concentration of 10mM and incubated at 37℃ for 10 minutes. After digestion, the digested lung tissue was homogenized using a gentle MACS (Miltenyi Biotec) and filtered through a 40 μm cell strainer, and the cells were collected.

Lung-infiltrating neutrophils were isolated using the Neutrophil Isolation Kit (Miltenyi Biotec) with a modification to enhance the depletion of non-neutrophil populations. Briefly, cell suspensions were incubated with biotin-conjugated anti-CD3ε (145-2C11) and anti-CD19 (6D5) antibodies prior to the labeling with anti-biotin Microbeads followed by depletion of non-neutrophils according to the manufacturer's protocol. The purity of isolated neutrophils was confirmed by flow cytometry and consistently exceeded 95%.

## Flow cytometric analysis

Cells were incubated with anti-CD16/32 antibody (Biolegend) for 15 minutes on ice to block non-specific Fc receptor binding, and then incubated with fluorochrome-conjugated following antibodies for 20 minutes at 4℃. CD11b (M1/70), CD19 (6D5), CX3CR1 (SA011F11), Ly6C (HK1.4), Ly6G (1A8), TCRβ (H57-597), TCRγδ (GL3), B220 (RA3-6B2), CD8 (53-6.7), CD49b (DX5), and NK1.1 (PK136) antibodies purchased from Biolegend, and CCR2 (475301), CD11c (HL3), CD40 (3/23), CD45 (30-F11), CD64 (X54-5/7.1), MHC class II (I-A/I-E) (M5/114.15.2), SiglecF (E50-2440), CD4 (RM4–5), and CD19 (1D3) antibodies purchased from BD Biosciences were used. Dead cells were excluded from the analysis by using a Live/Dead cell stain kit purchased from Invitrogen (eBioscience). Cells were then fixed with fixation buffer (Biolegend) for 20 minutes at room temperature and subsequently transferred out of the BSL3 facility. Data were acquired on a FACSAria Fusion (BD Biosciences) or LSRFortessa (BD Biosciences), and analyzed using FlowJo software (BD Biosciences). The gating strategy for each cell fraction is shown in S9 Fig.

## Histological analysis

Lung tissue was collected and fixed in 10% neutral buffered formalin. Fixed tissues were embedded in paraffin and sectioned at 4µm thickness, and stained with hematoxylin and eosin. Pathological evaluation was performed in a blind manner using three randomly selected 400 × fields per tissue, and scored according to the DAD histologic scoring systems. The DAD scores [28] were determined as follows: 1; absence of cellular sloughing and necrosis, 2; uncommon solitary cellular sloughing and necrosis, 3; multifocal cellular sloughing and necrosis with uncommon septal wall hyalinization, 4; multifocal (>75% of field) cellular sloughing and necrosis with common and/or prominent hyaline membranes. The average score across three fields was calculated for each sample. All samples were imaged using an all-in-one fluorescence microscope (BZ-X 710) (Keyence) and analyzed with NDP Viewer software (NDP.view2) (Hamamatsu).

## RNA isolation and Quantitative RT-PCR

Total RNA was extracted using TRIzol Reagent (Life technologies) and Directzol (Zymo Research) according to the manufacturer's protocol. cDNA was synthesized using SuperScript VILO (Invitrogen). Quantitative RT-PCR was performed using PowerUp SYBR Green Master Mix (Applied Biosystems) for the following genes: SARS-CoV-2 *Nucleocapsid phosphoprotein* (*N1, N2* region selected by Centers for Disease Control and Prevention) and *ORF1b*, *Ifna*, *Ifnb1*, *Il1b*, *Il12b*, *Il6*, *Tnf*, *Ccl5*, *Cxcl1*, *Cxcl2*, *Zc3h12a* (*Reg1*), *Tsc22d3*, *Nr2f2, Ax1, Muc4,* and *Fgfr1*, and *Hprt* as a control gene. Target gene expression was calculated by the comparative method for relative quantification after normalization to *Hprt* expression. Primer sequences for the quantitative RT-PCR are listed in the S2 Table.

## RNA-sequencing

Lung infiltrating neutrophils were isolated using the Neutrophil Isolation Kit (Miltenyi Biotec) as described above, and total RNA was isolated from neutrophils using TRIzol reagent (Life technologies) and Directzol (Zymo Research). For the library preparation, NEBNext Ultra II Directional RNA Library Prep Kit for Illumina (NEB) was used and RNA-sequencing was performed on an Illumina NextSeq 500 system. Bioinformatic analysis of RNA-sequencing data was performed as described previously [58]. Briefly, the sequences were mapped to the mouse genome version GRCm39 using Hisat2 (version 2.1.0) [59]. Differential gene expression analyses were performed using DESeq2 package in R (version 1.20.0) [60] on read counts obtained by FeatureCounts (version 1.6.4) [61] were used for GSEA [62]. GSEA was performed with GSEA software (v.4.3.3) using the response to type I interferon mouse gene set (GO:0034340), the ranked list file, and the enrichment plot, normalized enrichment score (NES), and false discovery rate (FDR) were generated.

## ScRNA-sequencing

Lung infiltrating CD45$^+$ cells were isolated using the CD45 microbeads (Miltenyi Biotec). Samples collected from three mice for both WT or *Reg1*$^{+/-}$ mice were combined into one sample for subsequent processing in each time point after MA10 infection. The scRNA-seq analysis was performed using the Chromium Fixed RNA Profiling Kit (10X genomics) according to the manufacturer's instructions. Briefly, cells were fixed with 1 ml of fixation buffer and stored at 4℃ for 23.5 h. After removing the supernatant and adding 1 ml of quenching buffer, 100 μl of enhancer, and 275 μl of 50% glycerol, the Chromium Fixed RNA Profiling protocols were followed. For library preparation, Chromium Fixed RNA Kit, Mouse Transcriptome (10X genomics) was used and sequencing was performed on an Illumina NovaSeq 6000 system. Analysis of the resulting scRNA-seq data was conducted using the Seurat R package (version 5.0.1) [63]. Data of each sample was read in and 52,794 cells with more than 200 detected genes and less than 10% mitochondrial reads were retained. Cells were processed using a default workflow, including normalization (NormalizeData function), prediction of variable genes (FindVariableFeatures function), scaling (ScaleData), followed by dimensionality reduction using principal component analysis (PCA) and UMAP using the first 10 principal components (PCs). This revealed the presence of batch effects, which we removed using the Harmony R package (version 1.2.0), using the first 10 PCs as input [64]. The first 10 dimensions returned by Harmony were used for dimensionality reduction using UMAP, and clustering using functions FindNeighbors and FindClusters (using resolution = 1.2). This resulted in 22 clusters, which we annotated manually using the expression of known marker genes of several immune cell types. Marker genes used are listed in S3 Table.

For the further analysis of neutrophils, we first selected cells of clusters 9, 13, and 21 (Fig 3A). On these cells, we recalculated highly variable genes, ran PCA followed by Harmony and UMAP, and obtained 8 clusters of cells (S6A Fig). However, inspection of marker genes suggested that clusters 1, 6, 7, and 8 consisted of doublets of neutrophils and B, T, or NK cells. To confirm this, we generated 60 artificial doublet cells by merging the transcription data of 60 randomly sampled neutrophil cells (from clusters 9, 13, or 21) with 20 randomly sampled B cells (from cluster 2), 20 randomly sampled T cells (from clusters 1, 6, or 7), and 20 randomly sampled NK cells (from cluster 3). We added these 60 artificial doublets to the neutrophil cells and repeating the original processing steps, resulting in the UMAP plot of S6B Fig. Artificial doublets accumulated clearly inside 3 clusters here, while a large cluster contained relatively few artificial doublets, corresponding to the neutrophil clusters 2, 3, 4, and 5. Therefore, for the remainder of the neutrophil analysis, we focused on neutrophil clusters 2, 3, 4, and 5. Genes with differential expression between genotypes, clusters, or dpi were predicted using the FindMarkers function using default parameters.

## *In vivo* neutrophil depletion experiments

Ultra-LEAF purified anti Ly6G (clone 1A8) was purchased from Biolegend. The procedure was performed as previously reported [65]. In brief, Anti-Ly6G (100 μg in 100 μl PBS) was injected intraperitoneally from 2 days prior to MA10 infection to 4 dpi daily. Alternatively, an equivalent volume of PBS was injected as the control.

## Statistical analysis

Two-tailed Student's t-test was used for most statistical analyses (GraphPad Prism), and a *P* value < 0.05 was considered statistically significant. Body weight after MA10 infection between each group was analyzed by two-way ANOVA with Bonferroni's post-test.

## Supporting information

**S1 Fig. *Reg1*$^{+/-}$ mice show no significant alteration in the absolute number of lung immune cells. A)** The absolute numbers of the indicated cell types obtained from the lungs of 12-week-old female BALB/c WT and *Reg1*$^{+/-}$ mice (n = 3 per group) at a steady state. Absolute numbers were calculated from the number of cells counted after the single

cell suspension from the lung and the percentage of each cell type observed by flow cytometry. The bar graphs show the mean ± s.d. *P*-values are shown. (two-tailed unpaired Student's t-test) **B)** Representative H&E-stained lung sections (200 × magnification) from intranasally PBS administrated mice collected at 28 days after administration.
(EPS)

**S2 Fig. Cytokine and chemokine levels in WT and *Reg1*$^{+/-}$ mice following $10^2$ TCID$_{50}$ MA10 infection. A)** qPCR of *Ifnb1, Il12b, Tnf, Ccl5, Cxcl1* and *Cxcl2* mRNA expression in RNA extracts obtained from infected lungs at 3 and 6 dpi. The bar graphs show the mean ± s.d. (n = 4). *P*-values are shown. (two-tailed unpaired Student's t-test).
(EPS)

**S3 Fig. Under MA10 infection, the Ly6G antibody increased the proportion of Ly6C$^{high}$CCR2$^+$ monocytes. A)** Flow cytometry analysis of Ly6C$^{high}$CCR2$^+$ monocytes in CD45$^+$ live cells isolated from infected lung at 5 dpi.
(EPS)

**S4 Fig. *Reg1*$^{+/-}$ mice exhibit significant amelioration of pneumonia in $10^3$TCID$_{50}$ MA10 infection. A)** The absolute numbers of the indicated cell types infiltrating infected lungs on 3 dpi (n = 3 per group) and 6 dpi (n = 3 for WT, n = 4 for *Reg1*$^{+/-}$) by flow cytometry. **B)** qPCR of MA10 viral nucleocapsid phosphoprotein (N1, N2 region) and ORF1b RNA expression in lungs of infected mice collected 3 dpi (n = 3 per group) and 6 dpi (n = 3 for WT, n = 4 for Reg1$^{+/-}$). **C)** qPCR of *Ifna, Il1b* and *Il6* mRNA expression in RNA extracts obtained from infected lungs at 3 dpi (n = 3 per group) and 6 dpi (n = 3 for WT, n = 4 for *Reg1*$^{+/-}$). The bar graphs **(A-C)** show the mean ± s.d. (n = 3 or 4). *P*-values are shown. (two-tailed unpaired Student's t-test).
(EPS)

**S5 Fig. scRNA-seq revealed that, among innate immune cells in the lungs following infection with $10^3$ TCID$_{50}$ MA10, neutrophils highly express *Reg1*. A)** The expression of *Reg1* in the indicated lung cell types on 3 and 5 dpi in WT mice, as shown in Fig 3B, is presented.
(EPS)

**S6 Fig. scRNA-seq analysis identified doublets of neutrophils and other cell types. A)** UMAP representation of lung infiltrating neutrophil clusters (cluster 9, 13 and 21 in Fig 3A). **B)** Neutrophils plotted in Fig 3A as clusters 9, 13, and 21 in UMAP were extracted and further classified into 8 clusters as shown in **(A)**. Artificial doublets of 60 T cells (or B cells, or NK cells) and neutrophils were created and plotted in UMAP. The results revealed that clusters 2, 3, 4, and 5 consisted of true neutrophils, while the remaining clusters consisted of doublets. **C)** Dot plot of neutrophil developmental status enhancing genes in the indicated neutrophil clusters (UMAP from Fig 3C). **D)** Dot plot of neutrophil developmental status related transcriptional factors in the indicated neutrophil clusters (UMAP from Fig 3C).
(EPS)

**S7 Fig. scRNA-seq reveals that, at 3 dpi, cluster 3 neutrophils in *Reg1*$^{+/-}$ mice express lower level of ISGs than WT mice. A)** Heatmap of the gene sets whose expression was found to be upregulated after MA10 infection in Cluster 3 neutrophils. Gene expression is further segmented at the indicated time points (0dpi, 3dpi and 5dpi) obtained from WT and *Reg1*$^{+/-}$ mice.
(EPS)

**S8 Fig. Expression levels of candidate Reg1-target genes in *Reg1*$^{+/-}$ lung neutrophils. A)** qPCR of *Nr2f2, Axl, Muc4*, and *Fgfr1* mRNA expression in RNA extracts of lung infiltrating neutrophils at 0 dpi obtained from WT and *Reg1*$^{+/-}$ mice (n = 3 per group). The bar graphs show the mean ± s.d. *P*-values are shown. (two-tailed unpaired Student's t-test).
(EPS)

**S9 Fig. Gating strategies for flow cytometry analysis. A, B)** Gating strategies used to analyze alveolar macrophages, dendritic cells, Ly6C⁻CCR2⁻ monocytes, Ly6C⁺CCR2⁻ monocytes, Ly6C$^{high}$CCR2⁺ monocytes, interstitial macrophages, neutrophils, eosinophils, B cells, CD4⁺ T cells, CD8⁺ T cells and NK cells from BALB/c WT or *Reg1$^{+/-}$* mice presented on Fig 1E. The same strategy was used to analyze those cells in MA10 infected BALB/c WT or *Reg1$^{+/-}$* mice presented on Figs 2A, 2C, 5C, S1A and S4A.
(EPS)

**S1 Table. List of differentially expressed genes in the *Reg1$^{+/-}$* vs. WT comparison across various cell types identified by scRNA-seq.** Based on the scRNA-seq results, we summed the total reads for each cell type across 0, 3 and 5 dpi and compared gene expression between WT and *Reg1$^{+/-}$* mice. The table lists genes that were upregulated in *Reg1$^{+/-}$* mice, ranked by fold change. T cells represent the sum of clusters 1, 6, 7, and 12 in Fig 3A. B cells represent the sum of clusters 2 and 22 in Fig 3A. NK cells represent cluster 3 in Fig 3A. Monocytes/macrophages are the sum of clusters 4, 8, 11, 15 and 20 in Fig 3A. Alveolar macrophages represent the sum of clusters 5 and 14 in Fig 3A. pDCs represent cluster 10 in Fig 3A.
(XLSX)

**S2 Table. List of primers used for qPCR.** List of primers for qPCR in Figs 1F, 1G, 2A, 5B, 5C, S2A, S4B, S4C and S8A.
(EPS)

**S3 Table. List of marker genes used for the immune cell type annotation.** List of marker genes for the immune cell type annotation in Figs 3A, 3B, 3C, 3D, 4A, 4B, 4C, 4D, S5A, S5B, S6A, S6B, S6C, S6D and S7A.
(EPS)

**S1 Data. All raw values for all graphs represented in figures.**
(XLSX)

## Acknowledgments

We thank Yoshimi Okumoto for secretarial assistance. We also thank Kazuhiko Azuma and Syota Ohki (Department of Pathology, Chiba University) for their help in preparing HE-stained specimens from lung tissue collected in mouse infection experiments. We thank Yoshihiro Yoshitake and Yukari Sando (NGS core facility of the Graduate Schools of Biostudies, Kyoto University) for assistance with RNA-seq analysis. We also thank Yoshiaki Yasumizu (Department of Experimental Immunology, Osaka University) for helpful suggestions regarding the scRNA-seq experiments. We also thank Hiroyuki Wakaguri and Yutaka Suzuki (Department of Computational Biology and Medical Sciences, Graduate School of Frontier Sciences, The University of Tokyo) for the submission of scRNA-seq and RNA-seq datasets to the DNA Data Bank of Japan. We thank Koto Ishii and Keitaro Kawano (Kyoto University) for the assistance with some of the experiments and analysis. We thank the Division of Advanced Biomedicine, Medical Research Support Center, Graduate School of Medicine, Kyoto University for the use of the FACSAria Fusion (BD Biosciences). We thank the Medical Research Support Center, Graduate School of Medicine for the use of the LSRFortessa (BD Biosciences) and Bioanalyzer (Agilent Technologies).

## Author contributions

**Conceptualization:** Keiko Yasuda, Osamu Takeuchi.

**Data curation:** Keiko Yasuda, Junichi Aoki, Kotaro Tanaka, Shintaro Shichinohe, Chikako Ono, Yukiko Muramoto, Daisuke Motooka,Yuzuru Ikehara, Tokiko Watanabe.

**Formal analysis:** Keiko Yasuda, Junichi Aoki, Kotaro Tanaka, Alexis Vandenbon, Daiya Ohara, Songling Li, Keiji Hirota, Daron M. Standley.

**Resources:** Hitomi Watanabe, Gen Kondoh.

**Funding acquisition:** Keiko Yasuda, Kotaro Tanaka, Shintaro Shichinohe, Tokiko Watanabe, Osamu Takeuchi.

**Supervision:** Osamu Takeuchi.

**Validation:** Keiko Yasuda, Junichi Aoki, Kotaro Tanaka, Osamu Takeuchi.

**Writing – original draft:** Keiko Yasuda, Junichi Aoki, Kotaro Tanaka, Osamu Takeuchi.

**Writing – review & editing:** Keiko Yasuda, Junichi Aoki, Kotaro Tanaka, Osamu Takeuchi.

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
