## [Decision Letter · Decision Letter 0]

18 Nov 2025

Regnase-1-mediated regulation of neutrophils modulates SARS-CoV-2 pneumonia

PLOS Pathogens

Dear Dr. Takeuchi,

Thank you for submitting your manuscript to PLOS Pathogens. After careful consideration, we feel that it has merit but does not fully meet PLOS Pathogens's publication criteria as it currently stands. Therefore, we invite you to submit a revised version of the manuscript that addresses the points raised during the review process.

We look forward to receiving your revised manuscript.

Kind regards,

Shashank Tripathi

Guest Editor

PLOS Pathogens

Ashley St. John

Section Editor

Editor-in-Chief

PLOS Pathogens

orcid.org/0000-0003-2946-9497

Michael Malim

PLOS Pathogens

orcid.org/0000-0002-7699-2064

**Journal Requirements:**

1) Please provide an Author Summary. This should appear in your manuscript between the Abstract (if applicable) and the Introduction, and should be 150-200 words long. The aim should be to make your findings accessible to a wide audience that includes both scientists and non-scientists. Sample summaries can be found on our website under Submission Guidelines:

https://journals.plos.org/plospathogens/s/submission-guidelines#loc-parts-of-a-submission

2) Some material included in your submission may be copyrighted. According to PLOSu2019s copyright policy, authors who use figures or other material (e.g., graphics, clipart, maps) from another author or copyright holder must demonstrate or obtain permission to publish this material under the Creative Commons Attribution 4.0 International (CC BY 4.0) License used by PLOS journals. Please closely review the details of PLOSu2019s copyright requirements here: PLOS Licenses and Copyright. If you need to request permissions from a copyright holder, you may use PLOS's Copyright Content Permission form.

Potential Copyright Issues:

i) Figure 2B. Please confirm whether you drew the images / clip-art within the figure panels by hand. If you did not draw the images, please provide (a) a link to the source of the images or icons and their license / terms of use; or (b) written permission from the copyright holder to publish the images or icons under our CC BY 4.0 license. Alternatively, you may replace the images with open source alternatives. See these open source resources you may use to replace images / clip-art:

3) In the online submission form, you indicated that ScRNA-seq and RNA-seq data sets are available under the accession numbers DRR609866-DRR609879 in DNA Data Bank of Japan (http://ddbj.nig.ac.jp/DRASearch/) by checking we found no results. We strongly recommend all authors deposit their data before acceptance, as the process can be lengthy and hold up publication timelines. Please note that, though access restrictions are acceptable now, your entire minimal dataset will need to be made freely accessible if your manuscript is accepted for publication. This policy applies to all data except where public deposition would breach compliance with the protocol approved by your research ethics board. If you are unable to adhere to our open data policy, please kindly revise your statement to explain your reasoning and we will seek the editor's input on an exemption.

4) Please amend your detailed Financial Disclosure statement. This is published with the article. It must therefore be completed in full sentences and contain the exact wording you wish to be published.

2) If any authors received a salary from any of your funders, please state which authors and which funders..

**Reviewers' Comments:**

Reviewer's Responses to Questions

**Part I - Summary**

Reviewer #1: The manuscript ‘Regnase-1-mediated regulation of neutrophils modulates SARS-CoV-2 pneumonia’ focuses on the role of Regnase1 in SARS-CoV-2 associated lung inflammation, with a brief study on role of neutrophils conferring protection against SARS-CoV-2 infection in mice. The manuscript is easy to follow; however, the authors fail to address the link between neutrophil-mediated lung damage and protection against viral infection, which is already a big question in the field.The manuscript

Reviewer #2: This paper demonstrates that an RNase expressed in neutrophils, Regnase-1 can confer resistance to infection by a mouse-adapted SARS-CoV-2 MA-10 strain. The experimental results show that Regnase-1 haploinsufficient (Reg1⁺/⁻) mice show reduced pneumonia severity after infection with the MA10 strain as compared to wild-type mice. Also, neutrophils were found to be elevated in Reg1⁺/⁻ mice as compared to WT during MA-10 infection. Upon depleting the neutrophils both in Reg1⁺/⁻ and WT mice, severe lung inflammation was observed. From this, the authors implied that Reg1 expression in neutrophils had a crucial role in regulating the resistance to MA-10 infection. Further, scRNA-seq analysis revealed four different neutrophil-subsets during the course of MA-10 infection. One of these neutrophil-subsets characterized by strong type I interferon (IFN) signatures was decreased in Reg1⁺/⁻ mice, although there are other immune cells do show a significant role that the authors do not discuss.

With RNA-seq analysis of lung neutrophils the authors found that Reg-1 haploinsufficiency led to the upregulation of some genes associated with IFN-response signatures. One of them, Tsc22d3 exhibited significant upregulation in Reg-1 haploinsufficient neutrophils. Overexpression of Reg1 was found to suppress Tsc22d3 expression whereas a nuclease-inactive Regnase-1 failed to inhibit Tsc22d3 activity. Therefore, the findings in this paper suggest that Regnase-1 regulates neutrophil genes associated with ISG signatures by degrading certain mRNAs such as Tsc22d3. The authors also proposed that the targeted suppression of Reg1 could provide a novel therapeutic approach for COVID-19 and other viral infections. Although, the study provides some interesting observations, the authors’ analysis of the data and conclusions are not convincing. Please see my comments below.

Reviewer #3: The manuscript by Yasuda et al. uncovers how Regnase-1, involved in suppressing negative regulation of the IFN response genes, regulates neutrophils function during SARS-CoV-2 pneumonia. The authors compare the phenotype using BALB/c WT and Reg1 +/- mice and mouse-adapted SARS-CoV-2 (MA10) to investigate the role of Reg1. Reg1 +/- mice showed increased resistance to MA10 infection, exhibiting reduced body weight and inflammation with elevated neutrophil counts without any direct effect on virus gene copy number, which led them to hypothesize that neutrophils confer protection rather than directly clearing out virus infection. They validated the hypothesis by depleting the neutrophils using an anti-LY6 antibody, which resulted in increased infection and inflammation. Surprisingly, the transcriptome analysis revealed no significant differences in T cell, B cell, and NK cells between WT and Reg1 +/-. That is in coordination with comparable viral RNA levels and proinflammatory cytokines, suggesting that proinflammatory cytokine production is not the driving factor of the virus clearance. scRNAseq data revealed that cluster 3 and 4 increased at 5 dpi in both WT and Reg1 +/-; however, cluster 5 is diminished at both 3 and 5 dpi in the Reg1 haploinsufficiency model. The presence of mature neutrophil markers and absence of immature ones in both WT and Reg1 +/- suggested functional heterogeneity among the mature neutrophils rather than stages of maturation. The upregulation of proinflammatory cytokines, coupled with reduced ISGs and type I IFN-associated genes in the Reg1 +/- mice, suggests that Reg1 haploinsufficiency attenuates type I IFN response, which contributes to reduced disease severity observed in the Reg1 +/- mice. Finally, they attribute this phenotype to the upregulation of Tsc22d3, a negative regulator of Type I IFN, in the Reg1 +/- mice. The Reg1 enzymatically suppresses the Tsc22d3, particularly using the D141 residue, as D141N loses the ability to suppress the Tsc22d3.

**Part II – Major Issues: Key Experiments Required for Acceptance**

Reviewer #1: 1. Neutrophils infiltration and NETs are hallmark of lung injury in SARS-CoV-2 (and respiratory virus) infection. The authors show that depletion of neutrophils reduces lung damage (DAD scores) while Regnase1 haploinsufficiency leads to more damage due to neutrophils infiltration. They also claim that neutrophils are protective against SARS-CoV-2 infection, with only body-weight changes, without showing replicating virus (PFU) in lungs. There is a missing link between the two phenotypes. The virus itself can cause lung damage in the absence of neutrophils, which is addressed nowhere in the study

2. It is unclear why the authors decided to choose mouse-adapted SARS-CoV-2 strain, while the naturally occurring strains can also infect laboratory mice. It will add relevance if they use at least 1-2 strains (B.1.351 and/or emerging omicron strains) to replicate their findings.

3. The authors should explain why they use a Regnase1 haploinsufficient (+/-) model and not a completely deficient (-/-) mouse model. Why C57Bl6 background was not used and instead BALB/c was preferred? Genotyping data is missing. Does the phenotype enhance in Regnase1-/- mice?

4. Several quality controls are missing. E.g. Fig 1A/B/C; Fig 2D/E The authors should include mock-infected WT and Reg1+/- controls. This should partially answer comment 1 above, by comparing infected versus uninfected within WT mice.

5. The authors should also test the expression levels of Regnase1 in different cell types upon viral infection.

6. The authors could have used a control like Remdesivir in their assays.

7. Regnase1 can also degrade viral RNA, the impact of which was neither shown nor discussed, including in airway epithelial cells.

8. The LD50 data of MA strain is missing which could be added in the supplementary figure.

9. The authors reported increased alveolar macrophages upon infection in Regnase1+/- mice. Though the study focuses on neutrophils, the authors should at least briefly discuss the role of alveolar macrophages and their Regnase1-dependent functions.

10. To claim mice recovery post-infection, the assay should be extended beyond 5dpi in Fig 2D, similar to Fig 1A.

Reviewer #2: Major Comments:

The manuscript aims to describe the mechanisms of SARS-CoV-2 pathogenesis that contribute to disease severity (which is known in terms of neutrophil contribution to the disease severity). The authors tried to justify the mechanistic details with decent observations, in particular, in the regulatory role of Regnase-1 during MA-10 infection in mice. However, half of these observations lack proper statistical tests and indicate the authors conclusions biased. Additionally, the paper lacks appropriate references for the previous findings and methodologies mentioned. Some of the points have been listed below:

In Figure S1A, there is an evident difference in the number of B-cells, 〖CD4〗^+ T-cells and neutrophils between Reg1⁺/⁻ and WT mice which has been overlooked by the authors. It has been stated that there is no difference in the immune cell proportions amongst the Reg1⁺/⁻ and WT mice without including any statistics for the same.

In Figure 1E and Figure S2A, appropriate statistics should be included in order to comment about the observations made by the authors in the paper.

Similarly in Figure S3, the immune cell populations such as dendritic cells, monocytes, NK cells and 〖Ly6C〗^+ 〖CCR2〗^-monocytes differ in numbers between Reg1⁺/⁻ and WT mice but, the authors do not comment about this in the paper. There might be phenotypic changes due to the variation in these proportions which has not been noted by the authors.

The heatmap shown in Figure 4D, Ifitm3 shows an upregulation in Reg1⁺/⁻ mice as compared to the WT one (in the 5dpi panel) which is overlooked by the authors and, instead it has been generalised and mentioned that all the ISGs are downregulated (Line 264).

In Reg-1 haploinsufficient neutrophils, apart from Tsc22d3, Fgfr1 also shows similar levels of upregulation in its expression as compared to WT mice (Figure S6A). The statistics need to be included here, and possible downregulation of the other three genes shown in the figure should be commented upon by the authors.

Reviewer #3: (No Response)

**Part III – Minor Issues: Editorial and Data Presentation Modifications**

Reviewer #1: 1. Line 87: ‘contribute’ should be changed to ‘associated to’

2. Color codes missing in Fig 4B and D.

3. This is not the first report on roles of neutrophils in SARS-CoV-2 and other viral infections. The authors should change the sentence in line 364.

4. Some of the data points in Fig 5B (Reg1+/-) are shaded.

Reviewer #2: Minor Comments:

In the discussion of the paper (line 375), appropriate references need to be included for the previous findings been mentioned.

The anti-Lys6G method used for neutrophil depletion experiments shown in the paper should be described in more detail in the methodology section. The appropriate reference of the paper first reporting this protocol has not been cited by the authors.

Reviewer #3: 1. The authors rely mainly on viral RNA quantification (N1, N2, ORF1b) to infer comparable viral loads. Infectious titers (e.g., plaque assay or TCID₅₀) from lungs would provide stronger support for the claim that Reg1 haploinsufficiency protects independently of viral clearance.

2. Anti-Ly6G (clone 1A8) is widely used, but transient depletion can also affect some monocyte subsets. The authors should confirm depletion efficiency and specificity by flow cytometry and discuss possible off-target effects.

3. Fig 2E: The DAD score which is claimed as higher upon Ly6 treatment compared to untreated, is not statistically significant. Also, the animal numbers are not same across the groups (n=3 for the untreated WT and n=4 for the treated group)

4. Figure 1C and 2B-D: While both animal experiment conditions appear similar (3MLD50), WT untreated mice displayed 100% mortality at 5 dpi in Fig 1C; however, in Fig 2D animals began to gain weight at 5 dpi.

5. scRNA-seq pools three mice per condition/time point, treating each as one biological replicate. While practical, this limits statistical robustness. The authors should acknowledge this limitation and ideally include validation by qPCR or flow cytometry for key subset markers

PLOS authors have the option to publish the peer review history of their article (what does this mean? ). If published, this will include your full peer review and any attached files.

**Do you want your identity to be public for this peer review?** For information about this choice, including consent withdrawal, please see our Privacy Policy .

Reviewer #1: No

Reviewer #2: No

Reviewer #3: No

**Figure resubmission:**

**Reproducibility:**



---

## [Decision Letter · Decision Letter 1]

2 Feb 2026

Dear Prof Takeuchi,

We are pleased to inform you that your manuscript 'Regnase-1-mediated regulation of neutrophils modulates SARS-CoV-2 pneumonia' has been provisionally accepted for publication in PLOS Pathogens.

Best regards,

Shashank Tripathi

Guest Editor

PLOS Pathogens

Ashley St. John

Section Editor

PLOS Pathogens

Sumita Bhaduri-McIntosh

Editor-in-Chief

PLOS Pathogens

orcid.org/0000-0003-2946-9497

Michael Malim

Editor-in-Chief

PLOS Pathogens

orcid.org/0000-0002-7699-2064

Dear Dr. Takeuchi,

Thank you for submitting the revised version of your manuscript, to PLOS Pathogens. We have now evaluated your revision and the reviewers’ comments.

I am pleased to inform you that the manuscript is acceptable for publication in PLOS Pathogens, pending a final, minor update. Following the first revision, some minor but important comments raised by Reviewer 2 remain to be addressed. These comments do not affect the overall conclusions of the study, but we ask that you update the manuscript accordingly to ensure clarity and completeness.

Please revise the manuscript to address the remaining minor points from Reviewer 2 and resubmit the updated files at your earliest convenience. Once these changes are made, the manuscript will proceed directly to formal acceptance. No further round of peer review will be required.

We appreciate your careful attention to these final details and look forward to receiving your revised submission.

Thank you for choosing PLOS Pathogens for the publication of your work.

Sincerely,

Shashank Tripathi

Academic Editor

PLOS Pathogens

Reviewer Comments (if any, and for reference):

Reviewer's Responses to Questions

**Part I - Summary**

Reviewer #1: (No Response)

Reviewer #2: (No Response)

Reviewer #3: (No Response)

**Part II – Major Issues: Key Experiments Required for Acceptance**

Reviewer #1: (No Response)

Reviewer #2: The authors have tried to address all the major comments and suggestions provided for the manuscript. The datasets that were previously lacking appropriate statistical tests to support the author’s conclusions have been supplemented in all the relevant figures. However, the interpretations derived from these data should be articulated more clearly by the authors. For example, precise inferences are required regarding the differences in the numbers of dendritic cells, monocytes, or Ly6C⁺CCR2⁻ monocytes differences between Reg1+/– and WT mice as presented in Figure S4.

The discrepancy observed between the scRNA-seq and flow cytometry results for NK cells has been addressed by the authors as a technical issue related to the sample collection. The authors justification partly explain the discrepancy observed, however, it leaves an unresolved gap in the understanding of the functional role of Regnase-1 in NK cells during SARS-CoV-2 MA10 infection.

Although the authors suggest that Reg1 haploinsufficiency attenuates type I IFN responses in neutrophils during MA10 infection, some of the ISGs, such as Ifitm3, show an increase in Reg1+/– mice. The potential biochemical role of this phenotype can be explained further by the authors in the context of viral infection.

The authors have added appropriate statistical analysis to the revised Figure S8A where it is revealed that there is no significant difference in the expression levels of Fgfr1, Tsc22d3, Nr2f2, Axl, and Muc4 between WT and Reg1+/– neutrophils. A broader assessment of additional genes associated with IFN response signature should be considered before concluding about the suppression of these genes by Reg1 activity in neutrophils. Or indicate this as a limitation of the study.

Reviewer #3: (No Response)

**Part III – Minor Issues: Editorial and Data Presentation Modifications**

Reviewer #1: (No Response)

Reviewer #2: Overall, the authors have significantly addressed the major revision comments which has resulted in substantive improvements to the manuscript. But a more comprehensive and mechanistic explanation of the observed phenotypes would further strengthen the study and enhance the interpretations made by the authors.

Reviewer #3: (No Response)

PLOS authors have the option to publish the peer review history of their article (what does this mean? ). If published, this will include your full peer review and any attached files.

**Do you want your identity to be public for this peer review?** For information about this choice, including consent withdrawal, please see our Privacy Policy .

Reviewer #1: No

Reviewer #2: **Yes:** Sannula Kesavardhana

Reviewer #3: No

---

## [Editor Report · Acceptance letter]

Dear Prof Takeuchi,

We are delighted to inform you that your manuscript, "Regnase-1-mediated regulation of neutrophils modulates SARS-CoV-2 pneumonia," has been formally accepted for publication in PLOS Pathogens.

Best regards,

Sumita Bhaduri-McIntosh

Editor-in-Chief

PLOS Pathogens

orcid.org/0000-0003-2946-9497

Michael Malim

Editor-in-Chief

PLOS Pathogens

orcid.org/0000-0002-7699-2064